# Clinical outcomes of bortezomib-based therapy in Taiwanese patients with multiple myeloma: A nationwide population-based study and a single-institute analysis

Wei-Nung Liu[1], Chao-Feng Chang[2], Chi-Hsiang Chung[3,4], Wu-Chien Chien[3,5], Tzu-Chuan Huang[1], Yi-Ying Wu[1], Ching-Liang Ho[1], Jia-Hong Chen[1,6]*

1 Division of Hematology/Oncology, Department of Medicine, Tri-Service General Hospital, National Defense Medical Center, Taipei, Taiwan, 2 Division of Gastroenterology, Department of Medicine, Tri-Service General Hospital, National Defense Medical Center, Taipei, Taiwan, 3 School of Public Health, National Defense Medical Center, Taipei, Taiwan, 4 Taiwanese Injury Prevention and Safety Promotion Association, National Defense Medical Center, Taipei, Taiwan, 5 Department of Medical Research, Tri-Service General Hospital, National Defense Medical Center, Taipei, Taiwan, 6 Graduate Institute of Clinical Medicine, College of Medicine, Taipei Medical University, Taipei, Taiwan

* ndmc_tw.tw@yahoo.com.tw

**Data Availability Statement:** There are 2 database that was used in the work. The first database cannot be shared publicly because of National

## Abstract

### Purpose

In a retrospective cohort study, we report the current epidemiology of patients with multiple myeloma (MM) and analyze the real-world clinical outcomes of bortezomib-based therapy.

### Materials and methods

This retrospective study was mainly designed to evaluate the characteristics, treatment outcomes, and prognostic factors of patients with MM who received bortezomib-based therapy. We identified 5,726 patients in Taiwan with MM newly diagnosed between 2007 and 2015. Confidential data from the National Health Insurance Research Database (NHIRD) was used under strict guidelines, as it was made available in an electronic format for research purposes. In addition, we analyzed 96 patients who have been diagnosed with MM and were treated at the Tri-Service General Hospital (TSGH) between January 1, 2002, and December 31, 2018.

### Results

Patients receiving first-line treatment with bortezomib had longer overall survival (OS) compared to those who received non-first-line treatment (p<0.001). In addition, the statistically lowest risk of mortality was when patients received first-line bortezomib followed by an autologous hematopoietic stem cell transplant (adjusted hazard ratio = 0.39, p<0.001). In the TSGH study, the patients were enrolled between January 1, 2002, and December 31, 2018, with an initial diagnosis of MM; there were 96 individuals treated with bortezomib. There was no statistically significant difference in the OS or progression-free survival (PFS) according

Health Insurance Research Database (NHIRD), Taiwan. This data are available from the NHIRD (nhird@nhri.org.tw) for researchers who meet the criteria for access to confidential data. The second database contains single-institute data of TSGH and are within the manuscript and its Supporting Information files.

**Funding:** The authors received no specific funding for this work.

**Competing interests:** The authors have declared that no competing interests exist.

to the gender, myeloma type, International Staging System stage, or treatment regimen. There was a significant difference in the PFS in patients receiving first-line bortezomib treatment with transplantation compared to those without transplantation ($p = 0.021$).

## Conclusions

Bortezomib as a first-line treatment extended the OS in four-year mortality tracking and lowered the mortality risk according to the NHIRD analysis. In the TSGH analysis, the results indicated that the initial conditions of patients with MM have a lower influence on the OS and PFS after bortezomib-based therapy was administered.

## Introduction

Multiple myeloma (MM) is a malignant neoplasm of plasma cells that are naturally responsible for overproducing monoclonal antibodies [1, 2]. The cohort study of Huang [3] was the first report to comprehensively describe the epidemiology of MM in Chinese populations. Huang reported that the average incidence and mortality per 100,000 people were 0.75 and 0.59 from 1979 to 2003 [3]. In our previous study from 1997 to 2013, the average incidence per 100,000 people was 1.83, which is consistent with recent reports that indicated an increase in the incidence of MM in some Asian countries [3, 4]. Furthermore, the mortality decreased, accounting for an average of 0.44 per 100,000 deaths. Novel agents such as bortezomib and thalidomide were introduced to treat MM [5]. Prolonged progression-free survival (PFS) and overall survival (OS) were seen following the National Health Insurance (NHI) Bureau's support of thalidomide from 2009 and bortezomib from 2012 [6] and also as a result of advanced supportive care in recent years [7].

The Taiwanese NHI system, launched in 1995, currently covers 99% of the population of 23 million people [8]. In 1998, the NHI covered nearly 99% of the Taiwanese population. From 1997 to 2013, the NHI program inpatients comprised more than 15 million people. This nationwide database from Taiwan provides an opportunity to evaluate the epidemiology and survival outcomes of numerous patients with MM [9].

The use of the novel agent bortezomib was supported from 2012 by the Bureau of the NHI [9]. In this retrospective cohort study based in Taiwan, we report the current epidemiology of patients with MM and analyze the first-line versus the non-first-line effect of bortezomib-based therapy on the clinical outcomes of Taiwanese patients with MM. In addition, a single-institute analysis of patients with MM with first-line or non-first-line exposure to bortezomib-based therapy was conducted (Tri-Service General Hospital, TSGH).

## Materials and methods

### Data sources and study population

The National Health Research Institutes, under strict guidelines, makes all confidential patient data from the National Health Insurance Research Database (NHIRD) available in an electronic format for research purposes [4, 8, 10]. We recovered two data files: the NHIRD and the Longitudinal Health Insurance Database with all inpatient and outpatient records for cancer care [11]. We applied the codes of the International Classification of Diseases, 9th Revision, Clinical Modification (ICD-9-CM) to recover diagnosis information.

## Ethical considerations

The NHIRD provides encrypted personal patient information to maintain privacy and provides researchers with anonymous identification numbers associated with relevant claim information, including the patients' sex, dates of birth, medical services utilized, and prescriptions. Patient consent is not required for access to the NHIRD. The Institutional Review Board (IRB) of the TSGH approved this study (IRB No. 2-102-05-107). Our IRB specifically waived the consent requirement.

## Study participants in the NHIRD

Patient information for this study was provided by the NHIRD and required insurance approval; it included inpatient and outpatient cases. We identified 5,726 patients newly diagnosed with MM (ICD-9 code 203.0) from 2007 to 2015 as the MM cohort. The date of MM diagnosis was established as the index date for starting the measurement of follow-up person-years. All patients were followed up until death, censored for loss to follow-up, withdrawal from the insurance system, or until the end of 2015.

## Study participants in a single institute

A multiple clinical case study of patients newly diagnosed MM with exposure to bortezomib-based therapy was conducted. At the TSGH, 96 patients with an initial diagnosis of MM were enrolled between January 1, 2002, and December 31, 2018. Because the patients' records/information were anonymized and deidentified prior to analysis in this study, informed consent was not required. The study was performed under the guidelines of the Helsinki Declaration and approved by the Human Subjects Protection Offices (IRB) of the TSGH, National Defense Medical Center, Taiwan. All patients had symptomatic MM in accordance with the diagnostic criteria of the International Myeloma Working Group (IMWG) [12]. The clinical information collected from the medical records included age, gender, disease stage at diagnosis, type of myeloma, treatment regimen, hematopoietic stem cell transplantation (HSCT), follow-up duration, progression-free duration, survival status, and cause of death. The MM stage at diagnosis was in accordance with the International Staging System (ISS) [13]. Response to therapy and disease progression was defined according to the IMWG uniform response criteria [12].

## Statistical analysis

In the NHIRD analysis, the distributions of definite sociodemographic factors, including mortality in four-year tracking (with, without), gender (male, female), age, and autologous HSCT (with, without), were displayed in patients with MM undergoing bortezomib-based treatment. The study population was divided into two subgroups: the first was first-line use of bortezomib (the study group) and the other was non-first-line use (the comparison group). We calculated the hazard ratios (HRs) and the 95% confidence interval (CI) using the Cox proportional hazards model to assess the HR of the mortality risk in patients with MM. The multivariate Cox proportional hazards model was used to measure the mortality-association risk factors in patients with MM after adjustment for the timing of bortezomib treatment and sociodemographic characteristics. SAS version 9.1 (SAS Institute, Cary, NC, USA) was used for data analyzes; $p < 0.05$ indicated statistical significance.

In the single-institute study, the OS and PFS were estimated using the Kaplan–Meier (KM) method and the difference in survival between the subgroups was compared using the log-rank test. In order to investigate the associated factors of the OS and PFS, univariate and multivariate Cox proportional hazard regressions were performed. The results are presented as HRs and their corresponding 95% CIs. In the subgroup analysis, the patients' characteristics were

compared as in the NHIRD subgroup design (first-line use of bortezomib versus non-first-line use of bortezomib) using the chi-squared test. All data analyzes were performed using SPSS software version 25 (SPSS Inc., Chicago, IL, USA).

## Results

### NHIRD analysis

Fig 1 shows the flowchart of the study sample selection from the Taiwanese NHIRD. As a result, 5,580 patients with MM were enrolled. Among them, there were 1,116 patients with MM from January 2007 to October 2011 as the comparison cohort and 4,610 patients with MM from November 2011 to December 2015 as the study cohort with a 1 : 4 pairing by gender, age, and so forth. Both cohorts were followed up for four years. Table 1 shows the distribution of the basic characteristics of the patients with MM. There were no significant differences between the groups in terms of gender, age, concomitant diseases, autologous HSCT, and Charlson Comorbidity Index scores, except for mortality in four-year tracking ($p = 0.001$). The KM analysis of cumulative OS in the four-year tracking demonstrated that the study cohort had longer OS compared to the control cohort ($p < 0.001$, Fig 2).

The single-variable and multivariable analyses of the mortality risk factors are shown in Table 2. Non-first-line use of bortezomib, male gender, and older age were the significant factors for increased myeloma death in the four-year tracking. The influence of the Charlson Comorbidity Index_R (CCI_R) score and patients with/without autologous HSCT were only statistically important in the multivariable analysis (adjusted HR: 1.075, $p = 0.03$; adjusted HR: 0.73, $p = 0.04$, respectively). Furthermore, the importance of autologous HSCT in different subgroups was analyzed and is shown in Fig 3. Patients who received first-line bortezomib-containing regimens followed by autologous HSCT had better survival benefit compared to the other groups (adjusted HR: 0.39, $p<0.001$).

### Single-institute analysis

In addition to the nationwide population-based cohort study, a single-institute analysis of patients with MM and bortezomib-based therapy was performed. The patients' profiles are described in Table 3. The 96 patients with MM treated with bortezomib-based therapy comprised 51 males (51.7%) and 45 females (48.3%), with a median age of 68.7 ± 12 years. There were 13 patients with stage 1 disease (5.2%), 36 patients with stage 2 disease (34.5%), and 47 patients with stage 3 disease (60.3%). 44 patients were diagnosed with IgG-type myeloma (45.8%), 29 with light-chain-type myeloma (30.2%), 21 with IgA-type myeloma (21.9%), and 2 with nonsecretory-type myeloma (2.1%).

Bortezomib-based therapy as the first-line treatment had estimated OS of 56.4 months and PFS of 35 ± 6.5 months. There was no statistical difference in the OS or PFS according to gender, myeloma type, ISS stage, or treatment regimen. However, there was a significant difference in PFS between patients receiving first-line bortezomib with autologous HSCT compared to those receiving first-line bortezomib without HSCT (52.8 ± 8.0 months versus 31.5 ± 7.0 months, $P = 0.021$) (Fig 4).

When first-line bortezomib-based therapy was compared with non-first-line bortezomib therapy, there was no significant difference in the OS (56.4 versus 70.6 months, $P = 0.76$) (Fig 5).

## Discussion

Bortezomib is an anticancer drug and the first therapeutic proteasome inhibitor to be used in humans [14]. Bortezomib interrupts this process and enables normal mechanism to kill cancer

**Fig 1. Flowchart of the study sample selection from the NHIRD in Taiwan.** MM: ICD-9-CM 203.0. Inclusion criteria: MM. Exclusion criteria: gender unknown.

**Table 1. Characteristics of patients with MM treated with bortezomib in the NHIRD analysis.**

| Bortezomib | Total | | First-line | | Non-first-line | | P |
|---|---|---|---|---|---|---|---|
| Variables | *n* | % | *n* | % | *n* | % | |
| **Total** | 5,580 | | 4,464 | 80.00 | 1,116 | 20.00 | |
| **Mortality in four-year tracking** | | | | | | | 0.001* |
| Without | 4,657 | 83.46 | 3,762 | 84.27 | 895 | 80.20 | |
| With | 923 | 16.54 | 702 | 15.73 | 221 | 19.80 | |
| **Gender** | | | | | | | 0.999 |
| Male | 3,370 | 60.39 | 2,696 | 60.39 | 674 | 60.39 | |
| Female | 2,210 | 39.61 | 1,768 | 39.61 | 442 | 39.61 | |
| **Age (years)** | 66.36 ± 12.65 | | 66.20 ± 12.46 | | 67.01 ± 13.35 | | 0.056 |
| **DM** | | | | | | | 0.518 |
| Without | 4,975 | 89.16 | 3,974 | 89.02 | 1,001 | 89.70 | |
| With | 605 | 10.84 | 490 | 10.98 | 115 | 10.30 | |
| **HTN** | | | | | | | 0.953 |
| Without | 4,837 | 86.68 | 3,869 | 86.67 | 968 | 86.74 | |
| With | 743 | 13.32 | 595 | 13.33 | 148 | 13.26 | |
| **COPD** | | | | | | | 0.828 |
| Without | 5,308 | 95.13 | 4,245 | 95.09 | 1,063 | 95.25 | |
| With | 272 | 4.87 | 219 | 4.91 | 53 | 4.75 | |
| **CAD** | | | | | | | 0.559 |
| Without | 4,650 | 83.33 | 3,713 | 83.18 | 937 | 83.96 | |
| With | 930 | 16.67 | 751 | 16.82 | 179 | 16.04 | |
| **Stroke** | | | | | | | 0.604 |
| Without | 5,363 | 96.11 | 4,287 | 96.03 | 1,076 | 96.42 | |
| With | 217 | 3.89 | 177 | 3.97 | 40 | 3.58 | |
| **CCI_R** | 0.68 ± 1.01 | | 0.69 ± 1.01 | | 0.65 ± 0.99 | | 0.228 |
| **BMT** | | | | | | | 0.209 |
| Without | 4,712 | 84.44 | 3,756 | 84.14 | 956 | 85.66 | |
| With | 868 | 15.56 | 708 | 15.86 | 160 | 14.34 | |

P-value (categorical variables: chi-squared/Fisher's exact test; continuous variables: *t*-test).

CCI_R: Charlson Comorbidity Index excluding cancer.

DM: diabetes mellitus; HTN: hypertension; COPD: chronic obstructive pulmonary disease; CAD: coronary artery disease; MM: multiple myeloma; BMT: bone marrow autotransplantation.

*P < 0.05

cells. In May 2003, seven years after its initial synthesis, bortezomib was approved in the United States by the US Food and Drug Administration (FDA) for use in MM on the basis of the results of the SUMMIT Phase II trial [15]. Bortezomib was approved by the US FDA for the initial treatment of patients with MM in 2008 [16].

The use of novel agents such as bortezomib was supported as a first-line treatment from 2012 by the Bureau of the NHI, Taiwan, which prolonged the PFS and OS duration. During the past three decades, the incidence of MM has increased significantly in Taiwan. The possible causes may be improved diagnostic techniques, aging of the population, and increased exposure to carcinogens. In previous studies, the introduction of novel agents, such as thalidomide and bortezomib, as a first-line treatment for patients with MM resulted in a 24% decrease in fatality [17]. To the best of our knowledge, there is no published literature available that compares the efficacy of first-line to non-first-line bortezomib-based treatment in Taiwan.

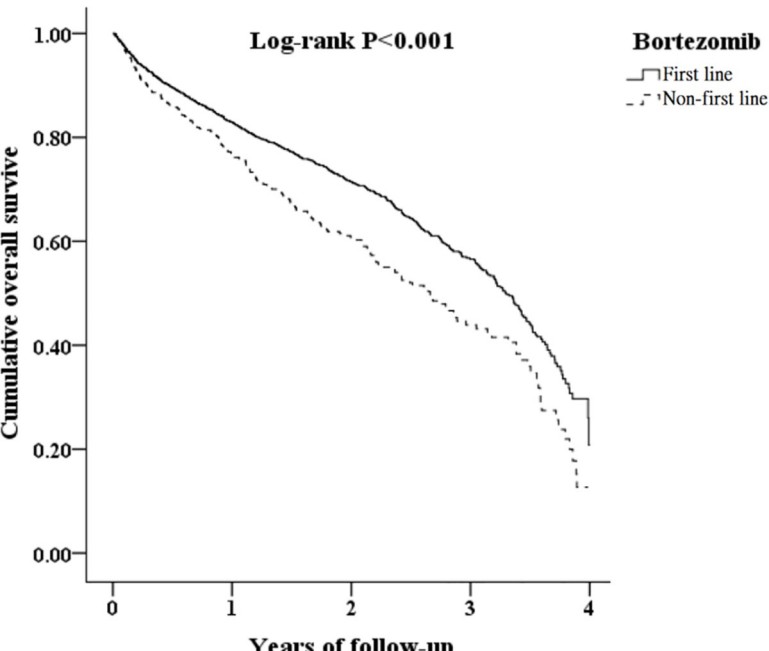

**Fig 2. KM test for cumulative OS in four-year tracking among patients with MM aged 18 and above treated with bortezomib (NHIRD analysis).**

In our NHIRD analysis, the risk of hospital death was significantly lower for the study cohort than for the control cohort in the four-year mortality tracking. Regarding the synergism of bortezomib use and autologous HSCT, the results showed that patients with MM receiving first-line bortezomib-based regimen and autologous HSCT treatment were the lowest hospital death risk group and that patients receiving non-first-line bortezomib without bone marrow transplantation were the highest risk group. This was consistent with the real-world experience [18]. Patients with MM who receive bone marrow transplantation are under 60 years of age [19]. For every one year added to the patient's age, the probability of hospital death increased by 1.7% (p < 0.001) and the probability of hospital death in men was 1.92 times that in women (p < 0.001).

In the subgroup ineligible for autologous HSCT, first-line bortezomib treatment did reduce the hospital death risk (the first-line was 0.681 times that of the non-first-line) (p < 0.001). In the non-first-line bortezomib treatment group, the hospital death risk was 0.724 times that of the nonautologous HSCT group (p = 0.040). Furthermore, the hospital death risk in the first-line bortezomib without HSCT group was similar to that in the non-first-line bortezomib with HSCT group (0.683 versus 0.725). In short, bortezomib as a first-line treatment was important and effective in reducing the hospital death risk for MM.

There were some limitations in the NHIRD analysis. First, the NHIRD did not offer detailed information, such as tumor stage and cause of death, so we could not show stage-stratified survival rates and disease-specific survival rates. Second, we were unable to have direct access to the patients to obtain additional information because of the anonymity ensured by the identification numbers. Third, we could not evaluate certain biological factors, such as genomics, because they were not included in the NHIRD. The staging system for MM includes serum albumin, B2 macroglobulin and cytogenetic profiles, and gene expression profiles. However, these data were limited in the NHIRD. Finally, the results derived from a cohort study are generally of a lower methodological quality than those from randomized trials,

**Table 2. Mortality factors in four-year tracking among patients with MM treated with bortezomib using Cox regression (NHIRD analysis).**

| Variables | Crude HR | 95% CI | 95% CI | P | Adjusted HR | 95% CI | 95% CI | P |
|---|---|---|---|---|---|---|---|---|
| **Bortezomib** | | | | | | | | |
| First-line | 0.709 | 0.609 | 0.825 | <0.001*** | **0.683** | 0.586 | 0.795 | <0.001*** |
| Non-first-line | Reference | | | | Reference | | | |
| **Gender** | | | | | | | | |
| Male | 1.292 | 1.129 | 1.478 | <0.001*** | **1.292** | 1.128 | 1.479 | <0.001*** |
| Female | Reference | | | | Reference | | | |
| **Age (years)** | 1.013 | 1.008 | 1.019 | <0.001*** | **1.017** | 1.011 | 1.023 | <0.001*** |
| **DM** | | | | | | | | |
| Without | Reference | | | | Reference | | | |
| With | 1.146 | 0.919 | 1.429 | 0.228 | 1.153 | 0.915 | 1.454 | 0.228 |
| **HTN** | | | | | | | | |
| Without | Reference | | | | Reference | | | |
| With | 1.368 | 1.107 | 1.692 | 0.004** | 1.078 | 0.710 | 1.637 | 0.724 |
| **COPD** | | | | | | | | |
| Without | Reference | | | | Reference | | | |
| With | 0.977 | 0.719 | 1.326 | 0.879 | 1.172 | 0.855 | 1.606 | 0.324 |
| **CAD** | | | | | | | | |
| Without | Reference | | | | Reference | | | |
| With | 1.351 | 1.118 | 1.683 | 0.002** | 1.355 | 0.934 | 1.965 | 0.110 |
| **Stroke** | | | | | | | | |
| Without | Reference | | | | Reference | | | |
| With | 0.987 | 0.710 | 1.371 | 0.936 | 1.152 | 0.823 | 1.612 | 0.410 |
| **CCI_R** | 1.055 | 0.991 | 1.123 | 0.092 | **1.075** | 1.006 | 1.149 | 0.033* |
| **BMT** | | | | | | | | |
| Without | Reference | | | | Reference | | | |
| With | 0.892 | 0.507 | 1.063 | 0.094 | **0.725** | 0.456 | 0.990 | 0.040* |

P-value (categorical variables: chi-squared/Fisher's exact test; continuous variables: *t*-test).

CCI_R = Charlson Comorbidity Index removed cancer

Crude HR: crude hazard ratio; CI: confidence interval; adjusted HR: adjusted variables listed in the table.

DM: diabetes mellitus; HTN: hypertension; COPD: chronic obstructive pulmonary disease; CAD: coronary artery disease; BMT: bone marrow autotransplantation;

MM: multiple myeloma

Nagelkerke's R-square of the model = 0.138.

* P < 0.05

** P < 0.01

*** P < 0.001.

because cohort study designs are subject to several biases related to the adjustment for confounders.

In order to overcome the limitations of the NHIRD analysis, a single-institute analysis was conducted for detailed prognostic factors and treatment outcome analysis. In the single-institute analysis, patients who were exposed to first-line bortezomib treatment showed no significant difference in the OS or PFS according to gender, myeloma type, ISS stage, and treatment response. The PFS was only significantly longer in the transplanted group compared to the nontransplanted group. This result means that once bortezomib is administered, patients can achieve a better clinical outcome consistent with previous studies [20–22].

The OS was not significantly different when first-line and non-first-line bortezomib were compared. This result was not consistent with the NHIRD analysis result, indicating that there

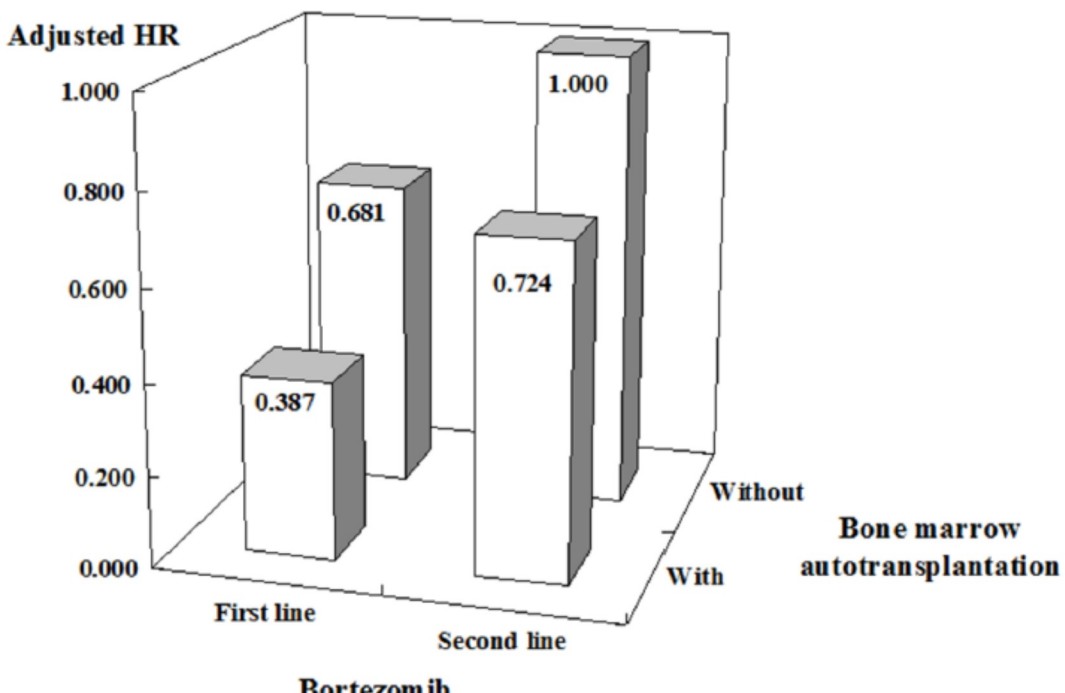

| Bortezomib | Bone marrow autotransplantation | Adjusted HR | 95% CI | P |
|---|---|---|---|---|
| First line | With | 0.387 | 0.111-0.524 | <0.001 |
| First line | Without | 0.681 | 0.582-0.789 | <0.001 |
| Second line | With | 0.724 | 0.453-0.987 | 0.035 |
| Second line | Without | Reference | | |

Adjusted HR (Hazard ratio): Adjusted variables listed in Table 2.

CI = confidence interval

Interaction term: Bortezomib × bone marrow autotransplantation, P=0.007

**Fig 3. Factors of mortality in four-year tracking among patients with MM stratified by bortezomib treatment and BMT (NHIRD analysis).**

may be some limitations among the enrolled patients. A review of the participants indicated there may be three limitations causing this result. First, the number of enrolled patients was smaller than in the NHIRD analysis (96 versus 5,580). Second, there were six patients with extremely long survival times in the non-first-line group. Third, there were eight patients enrolled in the first-line group who received a relatively short duration (less than three months) of bortezomib treatment. These limitations were mostly accounted for when performing the NHIRD analysis. However, the single-institute analysis still reflected the real-world bortezomib-based treatment experience and allowed more detailed influencing factors, such as myeloma type and response, to be analyzed.

In conclusion, first-line bortezomib-based therapy reduced the hospitalization death risk for Taiwanese patients with MM compared with non-first-line bortezomib-based therapy. Second, first-line bortezomib use in patients with MM who are ineligible for HSCT may have a clinical outcome similar to those eligible for HSCT. Third, the initial patient conditions play a less important role once the patients with MM have received bortezomib therapy.

**Table 3. Characteristics of patients with MM treated with bortezomib at a single institute (TSGH).**

| Bortezomib | Total | | First-line | | Non-first-line | | P |
|---|---|---|---|---|---|---|---|
| Variables | *n* | % | *n* | % | *n* | % | |
| Total | 96 | | 58 | 60.42 | 38 | 39.58 | |
| Mortality in tracking | | | | | | | 0.023 |
| Without | 44 | 45.83 | 32 | 55.17 | 12 | 31.58 | |
| With | 52 | 54.17 | 26 | 44.83 | 26 | 68.42 | |
| OS (months) | | | | 56.4 | | 70.6 ± 11.1 | 0.76 |
| Gender | | | | | | | 0.734 |
| Male | 51 | 53.13 | 30 | 51.72 | 21 | 55.26 | |
| Female | 45 | 46.88 | 28 | 48.28 | 17 | 44.74 | |
| Age (years) | 68.7 ± 12 | | 67.5 ± 11.9 | | 67.3 ± 11.2 | | 0.282 |
| Disease stage at diagnosis | | | | | | | 0.085 |
| I | 10 | 10.42 | 3 | 5.17 | 7 | 18.42 | |
| II | 34 | 35.42 | 20 | 34.48 | 14 | 36.84 | |
| III | 52 | 54.17 | 35 | 60.34 | 17 | 44.74 | |
| Myeloma type | | | | | | | 0.093 |
| IgG | 44 | 45.83 | 26 | 44.83 | 23 | 60.53 | |
| IgA | 21 | 21.88 | 13 | 22.41 | 6 | 15.79 | |
| Light chain | 29 | 30.21 | 19 | 32.76 | 7 | 18.42 | |
| Nonsecretory | 2 | 2.08 | 0 | 0 | 2 | 5.26 | |
| BMT | | | | | | | 0.092 |
| Without | 72 | 75 | 40 | 68.97 | 32 | 84.21 | |
| With | 24 | 25 | 18 | 31.03 | 6 | 15.79 | |

P-value (categorical variables: chi-squared/Fisher's exact test; continuous variables: *t*-test).

OS: overall survival; BMT: bone marrow autotransplantation; MM: multiple myeloma

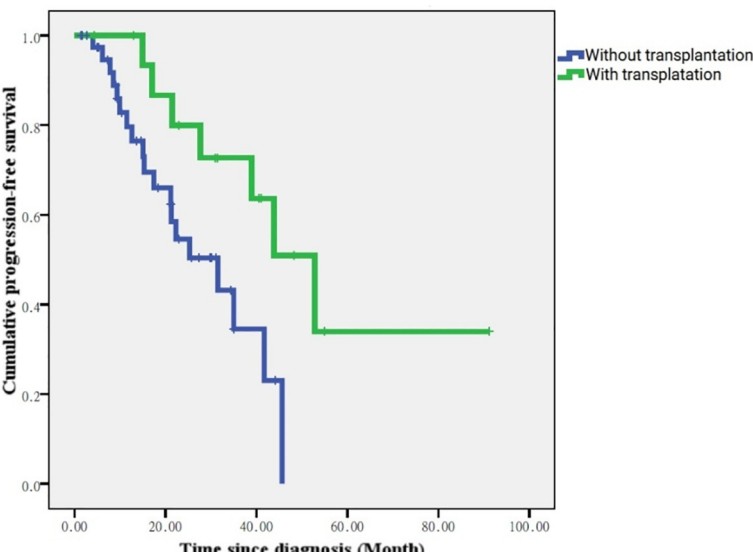

**Fig 4. KM test for PFS of bone marrow transplantation in the first-line bortezomib group (TSGH analysis).**

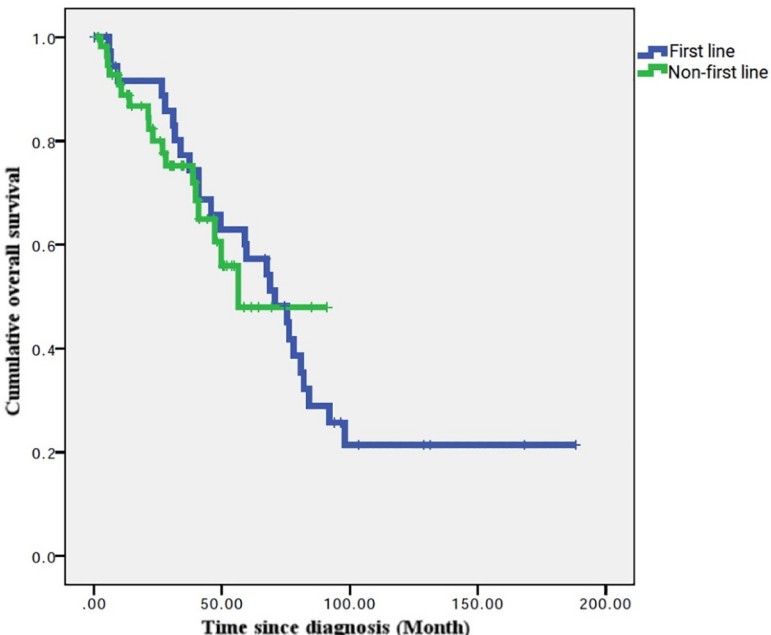

**Fig 5. KM test for cumulative OS in the first-line and non-first-line bortezomib group (TSGH analysis).**

In the future, we could prospectively collect nationwide data, including detailed information, to provide the most representative data on the bortezomib treatment results for patients with MM in Taiwan. This can not only provide stage-stratified survival and disease-specific survival rates but also help identify better clinical effects.

## Supporting information

**S1 File. The single-institute data of TSGH.**
(XLSX)

## Acknowledgments

The authors would like to thank the Cancer Registry Group of Tri-Service General Hospital for the clinical data support.

## Author Contributions

**Conceptualization:** Chi-Hsiang Chung.

**Data curation:** Wei-Nung Liu, Chao-Feng Chang, Chi-Hsiang Chung, Yi-Ying Wu.

**Formal analysis:** Wu-Chien Chien, Yi-Ying Wu.

**Investigation:** Ching-Liang Ho.

**Methodology:** Chi-Hsiang Chung, Wu-Chien Chien.

**Software:** Wu-Chien Chien.

**Supervision:** Tzu-Chuan Huang.

**Validation:** Wei-Nung Liu.

**Visualization:** Wei-Nung Liu.

**Writing – original draft:** Wei-Nung Liu.

**Writing – review & editing:** Tzu-Chuan Huang, Jia-Hong Chen.

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
