## [Decision Letter · Decision Letter 0]

31 Jul 2019

PONE-D-19-16145

The clinical outcomes of Bortezomib-based therapy in Taiwanese patients with multiple myeloma: A nationwide population-based study and a single institute analysis

PLOS ONE

Dear Dr Chen,

Thank you for submitting your manuscript to PLOS ONE. After careful consideration, we feel that it has merit but does not fully meet PLOS ONE’s publication criteria as it currently stands. Therefore, we invite you to submit a revised version of the manuscript that addresses the points raised during the review process.

Specifically, I agree that the manuscript should be extensively revised by an english-mother tongue scientific writer.

We would appreciate receiving your revised manuscript by Sep 14 2019 11:59PM. To enhance the reproducibility of your results, we recommend that if applicable you deposit your laboratory protocols in protocols.io, where a protocol can be assigned its own identifier (DOI) such that it can be cited independently in the future. For instructions see: http://journals.plos.org/plosone/s/submission-guidelines#loc-laboratory-protocols

We look forward to receiving your revised manuscript.

Kind regards,

Francesco Bertolini, MD, PhD

Academic Editor

PLOS ONE

2. Thank you for including your ethics statement:  "The NHIRD provides encrypted personal patient information to maintain privacy and provides researchers with anonymous identification numbers associated with relevant claim information, including patients’ sex, dates of birth, medical services utilized, and prescriptions. Patients’ consent is not required for accessing the NHIRD. The Institutional Review Board of TSGH approved this study. Our IRB specifically waived the consent requirement."  

a.Please amend your current ethics statement to include the full name of the ethics committee/institutional review board(s) that approved your specific study.

b.Once you have amended this/these statement(s) in the Methods section of the manuscript, please add the same text to the “Ethics Statement” field of the submission form (via “Edit Submission”).

3. In the Introduction section, we noticed several occurrences of overlapping text with your following previous publication, which need to be addressed:

https://journals.plos.org/plosone/article?id=10.1371/journal.pone.0167227

In your revision please ensure you quote or rephrase any duplicated text outside the methods section. Further consideration is dependent on these concerns being addressed.

4. We suggest you thoroughly copyedit your manuscript for language usage, spelling, and grammar. If you do not know anyone who can help you do this, you may wish to consider employing a professional scientific editing service.  

Reviewers' comments:

Reviewer's Responses to Questions

**Comments to the Author**

1. Is the manuscript technically sound, and do the data support the conclusions?

Reviewer #1: Yes

Reviewer #2: Yes

2. Has the statistical analysis been performed appropriately and rigorously? 

Reviewer #1: Yes

Reviewer #2: Yes

3. Have the authors made all data underlying the findings in their manuscript fully available?

Reviewer #1: Yes

Reviewer #2: Yes

4. Is the manuscript presented in an intelligible fashion and written in standard English?

Reviewer #1: No

Reviewer #2: Yes

5. Review Comments to the Author

Reviewer #1: This retrospective cohort study was designed to evaluate the characteristics, treatment outcome, and prognostic factors of MM patients who received Bortezomib-based therapy. The study was based on the Taiwan National Health Institute Research Database (NHIRD) and the Tri-Service General Hospital (TSGH).

This is an interesting epidemiological study. However, the study is difficult to follow due to poor English language skills.

Reviewer #2: In this work Liu and colleagues report the epidemiology of MM in Taiwan and analyze the outcome of bortezomib-treated MM patients. The study is based on data of Taiwan National Health Institute Research Database (NHIRD) and completed with a single institutional experience.

The work is well written and rationally conducted. Results are convincing and well discussed. The statistical analysis is rationale and adequate to achieve a correct evaluation of data. The study provide some useful information about epidemiology of MM in Taiwan and about the role of bortezomib therapy for the treatment of MM patients.

6. PLOS authors have the option to publish the peer review history of their article (what does this mean?). If published, this will include your full peer review and any attached files.

Reviewer #1: Yes: Carmelo Carlo-Stella

Reviewer #2: Yes: Roberto Ria

---

## [Author Response · Author response to Decision Letter 0]

13 Aug 2019

Dear Francesco Bertolini, Editor,

Thank you for inviting us to submit a revised draft of our manuscript entitled, “The clinical outcomes of Bortezomib-based therapy in Taiwanese patients with multiple myeloma: A nationwide population-based study and a single institute analysis” to PLOS ONE. We also appreciate the time and effort you and each of the reviewers have dedicated to providing insightful feedback on ways to strengthen our paper. Thus, it is with great pleasure that we resubmit our article for further consideration. We have incorporated changes that reflect the detailed suggestions you have graciously provided. We also hope that our edits and the responses we provide below satisfactorily address all the issues and concerns you and the reviewers have noted.

To facilitate your review of our revisions, the following is a point-by-point response to the questions and comments delivered in your letter dated 31 July 2019.

Question 1,

Answer,

Thank you for providing these insights. We had done our best to complete revision of the manuscript, the language, the reference form, and the terminology.

Question 2,

Thank you for including your ethics statement: "The NHIRD provides encrypted personal patient information to maintain privacy and provides researchers with anonymous identification numbers associated with relevant claim information, including patients’ sex, dates of birth, medical services utilized, and prescriptions. Patients’ consent is not required for accessing the NHIRD. The Institutional Review Board of TSGH approved this study. Our IRB specifically waived the consent requirement." 

Answer,

Thank you for your comments. The full name of the ethics committee/institutional review board(s) that approved our specific study was described in the new manuscript as “The Institutional Review Board (IRB) of the TSGH approved this study (IRB No. 2-102-05-107)”. ( Page 6, Line 4-5) We also added the same text to the “Ethics Statement” field of the submission form.

Question 3,

In the Introduction section, we noticed several occurrences of overlapping text with your following previous publication, which need to be addressed:

https://journals.plos.org/plosone/article?id=10.1371/journal.pone.0167227

In your revision please ensure you quote or rephrase any duplicated text outside the methods section. Further consideration is dependent on these concerns being addressed.

Answer,

Thank you for your comments. We had revised the overlapping text with our previous publication carefully. We also ensure to avoid duplicated text outside the methods section in the revision. 

Revision in manuscript:

Page 4, Line 11-13

In our previous study from 1997 to 2013, the average incidence per 100,000 people was 1.83, which is consistent with recent reports that indicated an increase in the incidence of MM in some Asian countries.

Page 4, Line 14-15

Furthermore, the mortality decreased, accounting for an average of 0.44 per 100,000 deaths. Novel agents such as bortezomib and thalidomide were introduced to treat MM.

Page 4, Line 16-19

Prolonged progression-free survival (PFS) and overall survival (OS) were seen following the National Health Insurance (NHI) Bureau’s support of thalidomide from 2009 and bortezomib from 2012 [6] and also as a result of advanced supportive care in recent years.

Page 4, Line 21-22

The Taiwanese NHI system, launched in 1995, currently covers 99% of the population of 23 million people.

Page 5, Line 3-9

The use of the novel agent bortezomib was supported from 2012 by the Bureau of the NHI. In this retrospective cohort study based in Taiwan, we report the current epidemiology of patients with MM and analyze the first-line versus the non-first-line effect of bortezomib-based therapy on the clinical outcomes of Taiwanese patients with MM. In addition, a single-institute analysis of patients with MM with first-line or non-first-line exposure to bortezomib-based therapy was conducted (Tri-Service General Hospital, TSGH).

Question 4,

We suggest you thoroughly copyedit your manuscript for language usage, spelling, and grammar. If you do not know anyone who can help you do this, you may wish to consider employing a professional scientific editing service.

Answer,

Thank you for your comments. We thoroughly copyedit our manuscript for language usage, spelling, and grammar by employing a professional scientific editing service. We also seriously add the certification in the resubmission. 

Responses to Reviewer's Questions

Reviewer #1: 

Comment: This retrospective cohort study was designed to evaluate the characteristics, treatment outcome, and prognostic factors of MM patients who received Bortezomib-based therapy. The study was based on the Taiwan National Health Institute Research Database (NHIRD) and the Tri-Service General Hospital (TSGH). This is an interesting epidemiological study. 

Question: The study is difficult to follow due to poor English language skills.

Response,

Thank you for your important comments. We had seriously and carefully copyedited our manuscript for language usage, spelling, and grammar by employing a professional scientific editing service. Our team will also do our best to improve English language skills to compose better article easy to follow in the future. 

Reviewer #2: 

Comment: In this work Liu and colleagues report the epidemiology of MM in Taiwan and analyze the outcome of bortezomib-treated MM patients. The study is based on data of Taiwan National Health Institute Research Database (NHIRD) and completed with a single institutional experience.

The work is well written and rationally conducted. Results are convincing and well discussed. The statistical analysis is rationale and adequate to achieve a correct evaluation of data. The study provide some useful information about epidemiology of MM in Taiwan and about the role of bortezomib therapy for the treatment of MM patients.

Response,

Thank you for your valuable comments. We will do our best to do valuable research in the future.

---

## [Decision Letter · Decision Letter 1]

3 Sep 2019

[EXSCINDED]

Clinical outcomes of bortezomib-based therapy in Taiwanese patients with multiple myeloma: A nationwide population-based study and a single-institute analysis

PONE-D-19-16145R1

Dear Dr. Chen,

We are pleased to inform you that your manuscript has been judged scientifically suitable for publication and will be formally accepted for publication once it complies with all outstanding technical requirements.

With kind regards,

Francesco Bertolini, MD, PhD

Academic Editor

PLOS ONE

Additional Editor Comments (optional):

Reviewers' comments:

Reviewer's Responses to Questions

**Comments to the Author**

1. If the authors have adequately addressed your comments raised in a previous round of review and you feel that this manuscript is now acceptable for publication, you may indicate that here to bypass the “Comments to the Author” section, enter your conflict of interest statement in the “Confidential to Editor” section, and submit your "Accept" recommendation.

Reviewer #1: All comments have been addressed

2. Is the manuscript technically sound, and do the data support the conclusions?

Reviewer #1: Yes

3. Has the statistical analysis been performed appropriately and rigorously? 

Reviewer #1: Yes

4. Have the authors made all data underlying the findings in their manuscript fully available?

Reviewer #1: Yes

5. Is the manuscript presented in an intelligible fashion and written in standard English?

Reviewer #1: Yes

6. Review Comments to the Author

Reviewer #1: The paper has been carefully revised according to the reviewer's comment with a specific emphasys on English language editing.

7. PLOS authors have the option to publish the peer review history of their article (what does this mean?). If published, this will include your full peer review and any attached files.

Reviewer #1: Yes: Carmelo Carlo-Stella

---

## [Editor Report · Acceptance letter]

10 Sep 2019

PONE-D-19-16145R1 

Clinical outcomes of bortezomib-based therapy in Taiwanese patients with multiple myeloma: A nationwide population-based study and a single-institute analysis 

Dear Dr. Chen:

I am pleased to inform you that your manuscript has been deemed suitable for publication in PLOS ONE. Congratulations! Your manuscript is now with our production department. 

With kind regards,

on behalf of

Dr. Francesco Bertolini 

Academic Editor

PLOS ONE